# Asgard archaea capable of anaerobic hydrocarbon cycling

Kiley W. Seitz[1], Nina Dombrowski [1,2], Laura Eme [3,4], Anja Spang[2,3], Jonathan Lombard[3], Jessica R. Sieber[5], Andreas P. Teske[6], Thijs J.G. Ettema [3,7] & Brett J. Baker[1]

Large reservoirs of natural gas in the oceanic subsurface sustain complex communities of anaerobic microbes, including archaeal lineages with potential to mediate oxidation of hydrocarbons such as methane and butane. Here we describe a previously unknown archaeal phylum, Helarchaeota, belonging to the Asgard superphylum and with the potential for hydrocarbon oxidation. We reconstruct Helarchaeota genomes from metagenomic data derived from hydrothermal deep-sea sediments in the hydrocarbon-rich Guaymas Basin. The genomes encode methyl-CoM reductase-like enzymes that are similar to those found in butane-oxidizing archaea, as well as several enzymes potentially involved in alkyl-CoA oxidation and the Wood-Ljungdahl pathway. We suggest that members of the Helarchaeota have the potential to activate and subsequently anaerobically oxidize hydrothermally generated short-chain hydrocarbons.

[1] Department of Marine Science, University of Texas Austin, Port Aransas, TX 78373, USA. [2] NIOZ, Royal Netherlands Institute for Sea Research, and Utrecht University, Den Burg 1797 SZ AB, The Netherlands. [3] Department of Cell and Molecular Biology, Science for Life Laboratory, Uppsala University, Uppsala SE-75123, Sweden. [4] Unité d'Ecologie, Systématique et Evolution, CNRS, Université Paris-Sud, Orsay 91400, France. [5] University of Minnesota Duluth, Duluth 55812 MN, USA. [6] Department of Marine Sciences, University of North Carolina, Chapel Hill 27599 NC, USA. [7] Laboratory of Microbiology, Department of Agrotechnology and Food Sciences, Wageningen University, Wageningen, NL-6708WE, The Netherlands. Correspondence and requests for materials should be addressed to B.J.B. (email: acidophile@gmail.com)

Short-chain alkanes, such as methane and butane, are abundant in marine sediments and play an important role in carbon cycling with methane concentrations of ~1 Gt being processed globally through anoxic microbial communities[1–3]. Until recently, archaeal methane cycling was thought to be limited to Euryarchaeota[4]. However, additional archaeal phyla, including Bathyarchaeota[5] and Verstraetarchaeota[6], have been shown to contain proteins with homology to the activating enzyme methyl-coenzyme M reductase (Mcr) and corresponding pathways for methane utilization. Furthermore, lineages within the Euryarchaeota belonging to Candidatus Syntrophoarchaeum spp., have been shown to use methyl-CoM reductase-like enzymes for anaerobic butane oxidation[7]. Similar to methane oxidation in many ANME-1 archaea, butane oxidation in Syntrophoarchaeum is proposed to be enabled through a syntrophic interaction with sulfur-reducing bacteria[7]. Metagenomic reconstructions of genomes recovered from deep-sea sediments from near 2000 m depth in Guaymas Basin (GB) in the Gulf of California have revealed the presence of additional uncharacterized alkyl methyl-CoM reductase-like enzymes in metagenome-assembled genomes within the Methanosarcinales (Gom-Arc1)[8]. GB is characterized by hydrothermal alterations that transform large amounts of organic carbon into methane, polycyclic aromatic hydrocarbons, low-molecular weight alkanes and organic acids allowing for diverse microbial communities to thrive (Supplementary Table 1)[8–11].

Recently, genomes of a clade of uncultured archaea, referred to as the Asgard superphylum that includes the closest archaeal relatives of eukaryotes, have been recovered from anoxic environments around the world[12–14]. Diversity surveys in anoxic marine sediments show that Asgard archaea appear to be globally distributed[12,14–16]. Based on phylogenomic analyses, Asgard archaea have been divided into four distinct phyla: Lokiarchaeota, Thorarchaeota, Odinarchaeota, and Heimdallarchaeota, with the latter possibly representing the closest relatives of eukaryotes[12]. Supporting their close relationship to eukaryotes, Asgard archaea possess a wide repertoire of proteins previously thought to be unique to eukaryotes known as eukaryotic signature proteins (ESPs)[17]. These ESPs include homologs of eukaryotic proteins, which in eukaryotes are involved in ubiquitin-mediated protein recycling, vesicle formation and trafficking, endosomal sorting complexes required for transport-mediated multivesicular body formation, as well as cytokinetic abscission and cytoskeleton formation[18]. Asgard archaea have been suggested to possess heterotrophic lifestyles and are proposed to play a role in carbon degradation in sediments; however, several members of the Asgard archaea also have genes that code for a complete Wood–Ljungdahl pathway and are therefore interesting with regard to carbon cycling in sediments[14,19].

Here, we present metagenome-assembled genomes (MAGs), recovered from GB deep-sea hydrothermal sediments, which represent an undescribed Asgard phylum with the metabolic potential to perform anaerobic hydrocarbon degradation using a methyl-CoM reductase-like homolog.

## Results

### Identification of Helarchaeota genomes from GB sediments.
We recently obtained more than ~280 gigabases of sequencing data from 11 samples taken from various sites and depths at GB hydrothermal vent sediments[20]. De novo assembly and binning of metagenomic contigs resulted in the reconstruction of over 550 genomes (>50% complete)[20]. these genomes we detected a surprising diversity of archaea, including >20 phyla, which appear to represent up to 50% of the total microbial community in some of these samples[20]. A preliminary phylogeny of the dataset using 37

concatenated ribosomal proteins revealed two draft genomic bins representing a previously unknown lineage of the Asgard archaea. These draft genomes, referred to as Hel_GB_A and Hel_GB_B, were re-assembled and re-binned resulting in final bins that were 82% and 87% complete and had a bin size of 3.54 and 3.84 Mbp, respectively (Table 1). An in-depth phylogenetic analysis consisting of 56 concatenated ribosomal proteins was used to confirm the placement of these final bins form a distant sister group with the Lokiarchaeota (Fig. 1a). Hel_GB_A percent abundance ranged from $3.41 \times 10^{-3}\%$ to $8.59 \times 10^{-5}\%$, and relative abundance from 8.43 to 0.212. Hel_GB_B percent abundance ranged from $1.20 \times 10^{-3}\%$ to $7.99 \times 10^{-5}\%$, and relative abundance from 3.41 to 0.22 compared to the total raw reads. For both Hel_GB_A and Hel_GB_B the highest abundance was seen at the site from which the bins were recovered. These numbers are comparable to other Asgard archaea whose genomes have been isolated form these sites[20]. Hel_GB_A and Hel_GB_B had a mean GC content of 35.4% and 28%, respectively, and were recovered from two distinct environmental samples, which share similar methane-supersaturated and strongly reducing geochemical conditions (concentrations of methane ranging from 2.3 to 3 mM, dissolved inorganic carbon ranging from 10.2 to 16.6 mM, sulfate near 21 mM and sulfide near 2 mM; Supplementary Table 1) but differed in temperature (28 and 10 °C, respectively, Supplementary Table 1)[21].

Phylogenetic analyses of a 16S rRNA gene sequence (1058 bp in length) belonging to Hel_GB_A confirmed that it is related to Lokiarchaeota and Thorarchaeota, but is phylogenetically distinct from either of these lineages (Fig. 1b). A comparison to published Asgard archaeal 16S rRNA gene sequences indicate a phylum level division between the Hel_GB_A sequence and other Asgard archaea with a percent identity of 82.67% when compared to Lokiarchaeum GC14_75[22] (Supplementary Table 2). A search for ESPs in both bins revealed that they contained a similar suite of homologs compared to those previously identified in Lokiarchaeota, which is consistent with their phylogenomic relationship (Fig. 2). Yet these lineages are relatively distantly related as evidenced by their difference in GC content and relatively low-pairwise sequence identity of proteins. An analysis of the average amino acid identity (AAI) showed that Hel_GB_A and Hel_GB_B shared 1477 genes and AAI of 51.96%. When compared to Lokiarchaeota_CR4, Hel_GB_A shares 634 orthologous genes out of 3595 and Hel_GB_B shares 624 out of 3157. Helarchaeota bins showed the highest AAI similarity to Odinarchaeota LCB_4 (45.9%); however, it contained fewer orthologous genes (574 out of 3595 and 555 out of 3157 for Hel_GB_A and Hel_GB_B, respectively). Additionally, the Hel_GB bins differed from Lokiarchaeota in their total gene number, for example Hel_GB_A possessed 3595 genes and CR_4 possessed 4218; this difference is consistent with the larger

### Table 1 Bin statistics for Helarchaeota bins

| SeqID | Hel_GB_A | Hel_GB_B |
|---|---|---|
| Completeness (%) | 82.4 | 86.92 |
| Contamination (%) | 2.8 | 1.40 |
| Strain heterogeneity (%) | 0 | 0 |
| Scaffold number | 333 | 182 |
| GC content (%) | 35.40 | 28.00 |
| N50 (bp) | 15,161 | 28,908 |
| Length total (Mbp) | 3.84 | 3.54 |
| Estimated Genome size (Mbp) | 4.6 | 4.1 |
| Longest contig (bp) | 52,512 | 72,379 |
| Mean contig (bp) | 11,531 | 19,467 |

Degree of completeness, contamination, and heterogeneity were determined using CheckM[40]

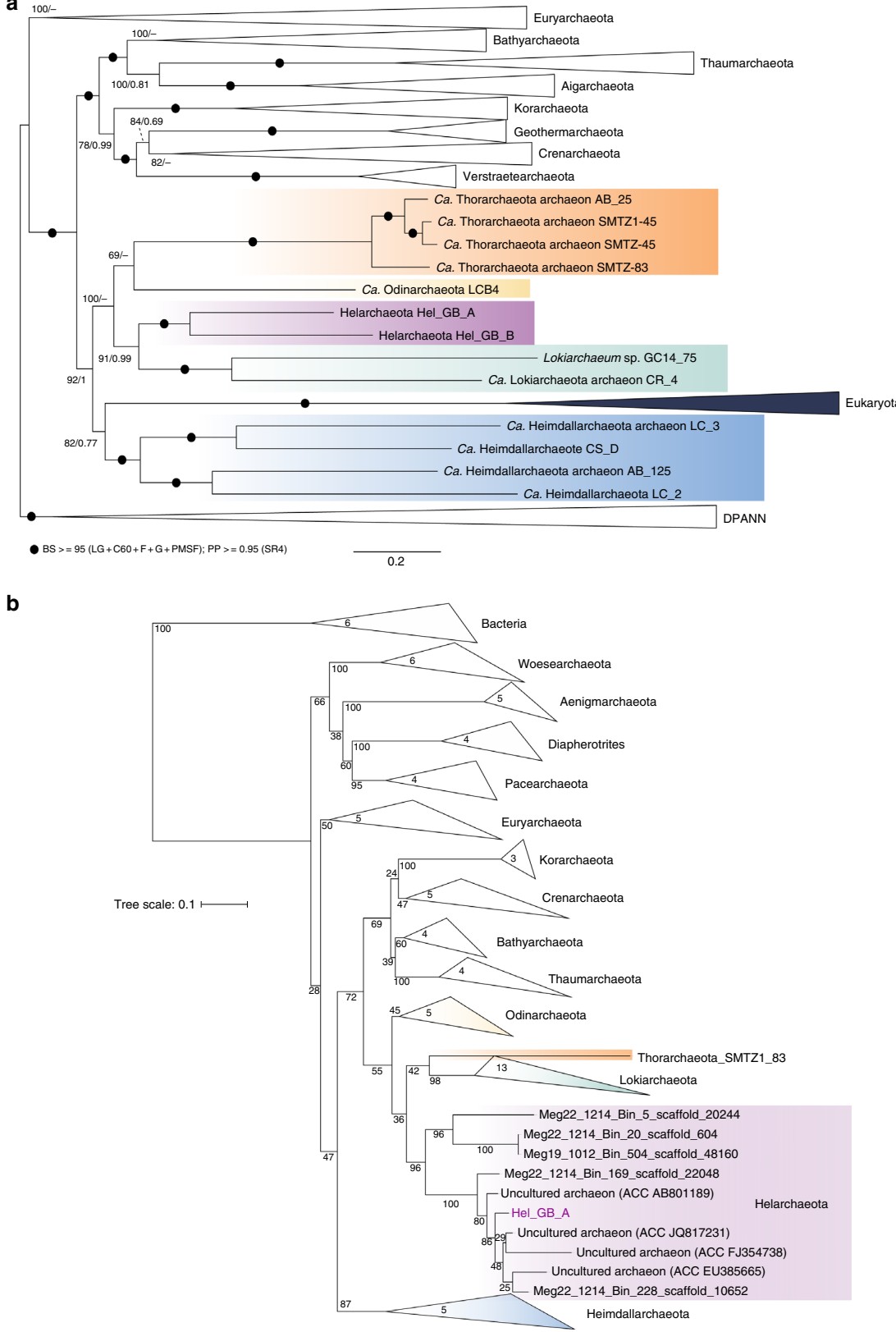

**Fig. 1** Phylogenomic position of Helarchaeota within the Asgard archaea superphylum. **a** Phylogenomic analysis of 56 concatenated ribosomal proteins identified in Helarchaeota bins. Black circles indicate bootstrap values greater than 95 (LG+C60+F+G+PMSF); posterior probability ≥ 0.95 (SR4). **b** Maximum-likelihood phylogenetic tree of 16S rRNA gene sequences thought to belong to Helarchaeota. The phylogeny was generated using RAxML (GTRGAMMA model and number of bootstraps determined using the extended majority-rule consensus tree criterion). The purple box shows possible Helarchaeota sequences from GB data, as well as closely related published sequences and sequences form recently identified Helarchaeota bins (identified as Megxx_xxxx_Bin_xxx_scaffold_xxxxx). Number of sequences is depicted in the collapsed clades

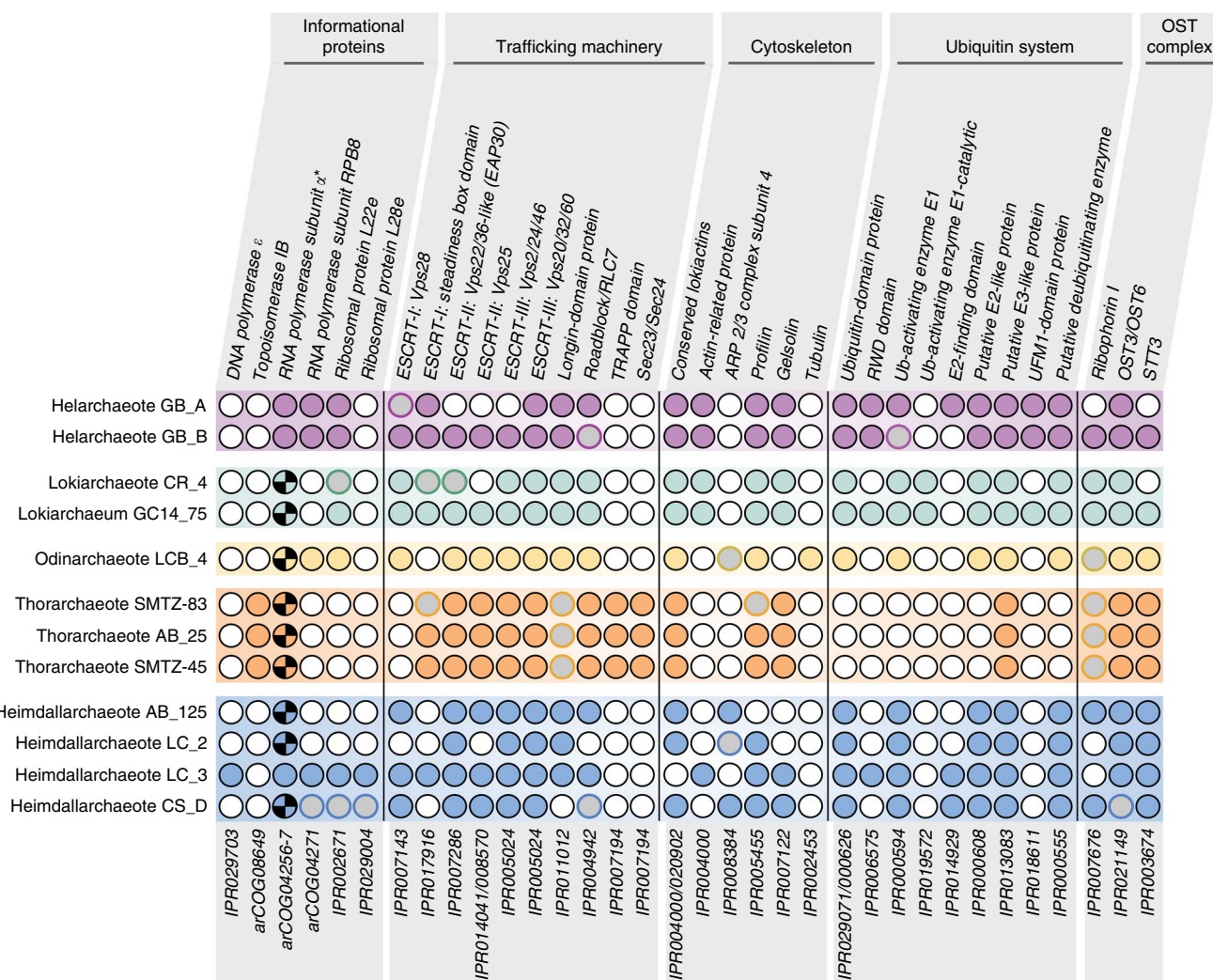

**Fig. 2** Distribution of eukaryotic signature proteins (ESPs) in Helarchaeota and other Asgard archaea. Numbers under each column correspond to the InterPro accession number (IPR) and Archaeal Clusters of Orthologous Genes (arcCOG) IDs that were searched for. Full circles refer to cases in which a homolog was found in the respective genomes. Empty circles with black outlines represent the absence of the ESP. The checkered pattern in the RNA polymerase subunit alpha represents the fact that the proteins were split, while the fused proteins are represented by the full circles. Gray circles with borders in any other color represent cases where the standard profiles were not found but potential homologs where detected. In the Roadblock proteins, potential homologs were detected, but the phylogeny could not support the close relationship of any of these copies to the Asgard archaea group closest to eukaryotes. In the Ub-activating enzyme E1 represents homologs found clustered appropriately with its potential orthologs in the phylogeny but the synteny of this gene with other ubiquitin-related proteins in the genome is uncertain

estimated genome size for Lokiarchaeum CR_4 compared to Hel_GB_A (~5.2 to ~4.6 Mbp) (Supplementary Table 3). These results add support to the phylum level distinction observed for Hel_GB_A and Hel_GB_B in both the ribosomal protein and 16S rRNA phylogenetic trees. We propose the name Helarchaeota after Hel, the Norse goddess of the underworld and Loki's daughter for this lineage.

**Metabolic analysis of Helarchaeota.** To reconstruct the metabolic potential of these archaea, the Helarchaeota proteomes were compared to several functional protein databases[20] (Fig. 3a). Like many archaea in marine sediments[23], Helarchaeota may be able to utilize organic carbon as they possess a variety of extracellular peptidases and carbohydrate degradation enzymes that include the β-glucosidase, α-L-arabinofuranosidase and putative rhamnosidase, among others (Supplementary Table 4 and Supplementary Data 1). Degraded organic substrates can then be metabolized via glycolysis and an incomplete TCA cycle from

citrate to malate and a partial gamma-aminobutyric acid shunt (Fig. 3a, Supplementary Data 1). Both Helarchaeota bins are missing fructose-1,6-bisphosphatase and have few genes coding for the pentose phosphate pathway. Genes encoding for the bifunctional enzyme 3-hexulose-6-phosphate synthase/6-phospho-3-hexuloisomerase (hps-phi) were identified in Hel_GB_B suggesting they may be using the ribulose monophosphate pathway for formaldehyde anabolism. Genes coding for acetate-CoA ligase (both APM and ADP-forming) and an alcohol dehydrogenase (adhE) were identified in both genomes suggesting that the organisms may be capable of both fermentation and production of acetyl-CoA using acetate and alcohols (Supplementary Data 1). Like in Thorarchaeota and Lokiarchaeota, these genomes possess the large subunit of type IV ribulose bisphosphate carboxylase[19,24]. In addition, the Helarchaeota genomes encode for the catalytic subunit of the methanogenic type III ribulose bisphosphate carboxylase used for C-fixation[24]. Helarchaeota are metabolically distinct from Lokiarchaeota as both Hel_GB draft genomes appear to lack a complete TCA cycle

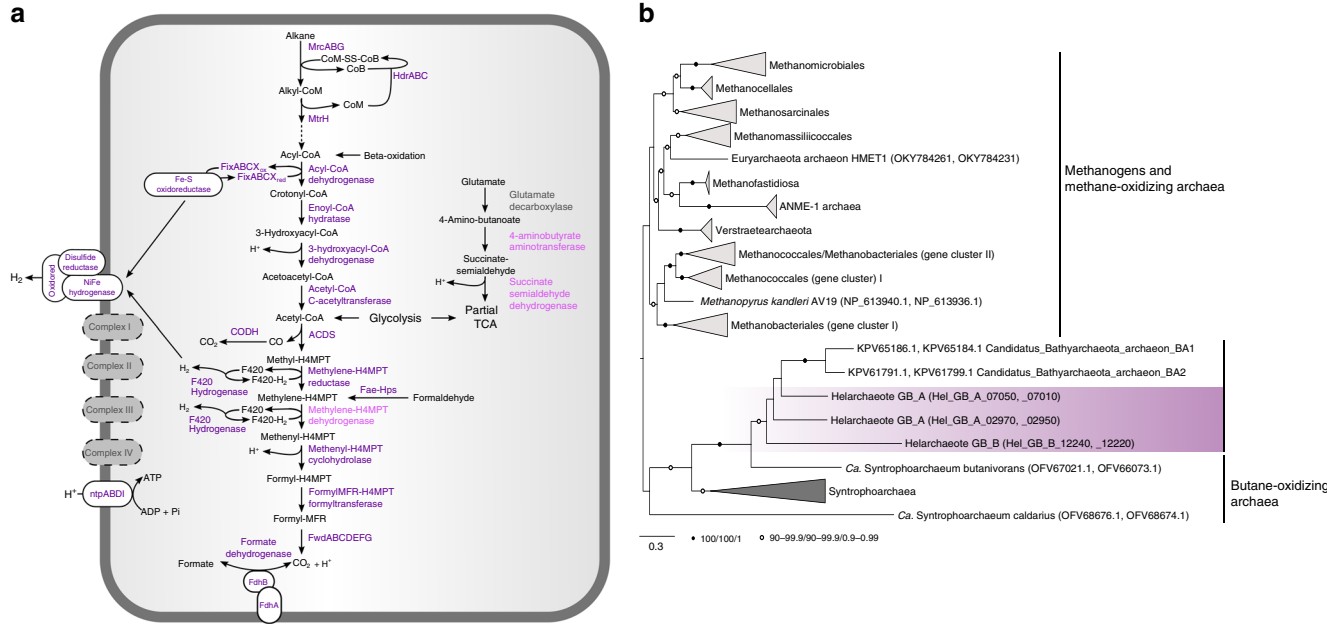

**Fig. 3** Metabolic inference of Helarchaeota and phylogenetic analyses of McrAB proteins. **a** Enzymes shown in dark purple are present in both genomes, those shown in light purple are present in a single genome and ones in gray are absent. **b** The tree of concatenated McrAB proteins was generated using IQ-tree with 1000 ultrafast bootstraps, single-branch test bootstraps and posterior probability values from the Bayesian phylogeny. White circles indicate support values of 90–99.9/90–99.9/0.9–0.99 and black filled circles indicate values of 100/100/1. The tree was rooted arbitrarily between the cluster comprising canonical McrAB homologs and divergent McrAB homologs. Scale bars indicate the average number of substitutions per site

as genes coding for citrate synthase and malate/lactate dehydrogenase are absent. Both genomes also likely produce acetyl-CoA using glyceraldehyde 3-phosphate dehydrogenase which is absent in Lokiarchaeota[19] (Supplementary Data 1). Helarchaeota genomes lack genes that code for enzymes involved in dissimilatory nitrogen and sulfur metabolism. Assimilatory genes, including *sat, cysN, and cysC* were found in Hel_GB_B, however, these genes were not identified in Hel_GB_A. This absence may be indicative of species-specific characteristics or could be a results of genome incompleteness. Additional genomes of members of the Helarchaeota will help to fully understand the diversity of these pathways across the whole phylum.

Interestingly, both Helarchaeota genomes have *mcrABG*-containing gene clusters encoding putative methyl-CoM reductase-like enzymes (Fig. 3b, Supplementary Figure 1)[4,5,7]. Phylogenetic analyses of both the A subunit of methyl-CoM reductase-like enzymes (Supplementary Figure 2) as well as the concatenated A and B subunits (Fig. 3b) revealed that the Helarchaeota sequences are distinct from those involved in methanogenesis and methane oxidation but cluster with homologs from butane-oxidizing Syntrophoarchaea[7] and Bathyarchaeota with high-statistical support (rapid bootstrap support/single-branch test bootstrap support/posterior probability of 99.8/100/1; Fig. 3b) excluding the distant homolog of *Ca. Syntrophoarchaeum caldarius* (OFV68676). Analysis of the Helarchaeota mcrA alignment confirmed that amino acids present at their active sites are similar to those identified in Bathyarchaeota and Syntrophoarchaeum methyl-CoM reductase-like enzymes (Supplementary Figure 3). In Syntrophoarchaeum, the methyl-CoM reductase-like enzymes have been suggested to activate butane to butyl-CoM[7]. It is proposed that this process is then followed by the conversion of butyl-CoM to butyryl-CoA; however, the mechanism of this reaction is still unknown. Butyryl-CoA can then be oxidized to acetyl-CoA that can be further feed into the Wood–Ljungdahl pathway to produce $CO_2$[7]. While some n-butane is detected in GB sediments (usually below 10 μM), methane is the most abundant hydrocarbon

(Supplementary Table 1) followed by ethane and propane (often reaching the 100 μM range); thus, a spectrum of short-chain alkanes could potentially be metabolized by Helarchaeota[25].

**Proposed hydrocarbon degradation pathway for Helarchaeota.** Next, we searched for genes encoding enzymes potentially involved in hydrocarbon utilization pathways, including propane and butane oxidation. Along with the methyl-CoM reductase-like enzyme that could convert alkane to alkyl-CoM, Helarchaeota possess heterodisulfide reductase subunits ABC (*hdrABC*), which is needed to recycle the CoM and CoB heterodisulfides after this reaction occurs (Figs. 3 and 4)[7,8]. The conversion of alkyl-CoM to acyl-CoA is currently not understood in archaea capable of butane oxidation. Specialized alkyl-binding versions of methyl-transferases would be required to convert alkyl-CoM to butyl-CoA or other acyl-CoAs, as discussed for *Ca. S. butanivorans*[7]. Genes coding for methyltransferases were identified in both Helarchaeota genomes, including a likely tetra-hydromethanopterin S-methyltransferase subunit H (MtrH) homolog (Fig. 4; Supplementary Data 1). Short-chain acyl-CoA could be oxidized to acetyl-CoA using the beta-oxidation pathway via a short-chain acyl-CoA dehydrogenase, enoyl-CoA hydratase, 3-hydroxyacyl-CoA dehydrogenase, and acetyl-CoA acetyl-transferase, candidate enzymes for all of which are present in the Helarchaeota genomes and are also found in genomes of other Asgard archaea (Fig. 4)[19]. Along with these enzymes, genes coding for the associated electron transfer systems, including an Fe–S oxidoreductase and all subunits of the electron transfer flavoprotein complex were identified in Helarchaeota (Fig. 4). Acetyl-CoA produced by beta-oxidation might be further oxidized to $CO_2$ via the Wood–Ljungdahl pathway, using among others the classical 5,10-methylene-tetrahydromethanopterin reductase (Figs. 3a and 4).

**Possible energy-transferring mechanisms for Helarchaeota.** To make anaerobic alkane oxidation energetically favorable, it must

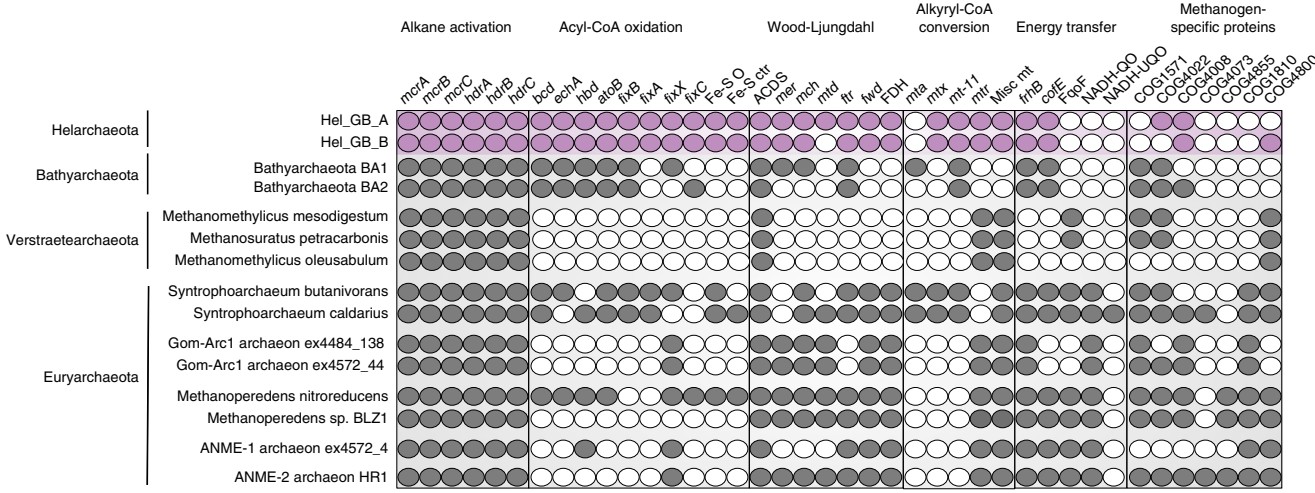

**Fig. 4** Comparison of Helarchaeota metabolism to other alkane oxidizing and methanogenic archaea. Alkane metabolism of Helarchaeota compared to Bathyarchaeota and *Ca*. Syntrophoarchaeum sp., Verstraetearchaeota, GoM-Arc1 sp., ANME-1 sp., and ANME-2 sp. A list of genes and corresponding contig identifiers can be found in Supplementary Data 1. Filled circles represent the presences of genes and white circles represents absence of genes

be coupled to the reduction of an internal electron acceptor or transferred to a syntrophic partner that can perform this reaction[7,26,27]. We could not identify an internal electron sink or any canonical terminal reductases used by ANME archaea (such as iron, sulfur, or nitrogen reductases), leading to the conclusion that a syntrophic partner organism would be necessary to enable growth on short-chain hydrocarbons. However, we could not identify any obvious syntrophic partner organisms based on co-occurrence analyses of abundance profiles in metagenomic datasets generated in this study[20].

An evaluation of traditional energy transferring mechanisms showed that the Helarchaeota bins lack genes coding for NADH: ubiquinone oxidoreductase, $F_{420}$-dependent oxidoreductase, $F_{420}H_2$:quinone oxidoreductase and NADH:quinone oxidoreductase that were identified in *Ca. S. butanivorans* (Fig. 4)[7]. These protein complexes are important for energy transfer across the cell membrane and are common among syntrophic organisms[2,28,29]. Helarchaeota also lack genes coding for pili or cytochromes that are often involved in direct electron transfer to a bacterial partner, as demonstrated for different ANME archaea[26,30]. Therefore, Helarchaeota may use a thus far unknown mechanism for energy conservation. Below we analyzed potential energy-transferring mechanisms that might be involved in syntrophic interactions between Helarchaeota and potential partner organisms.

A possible candidate for energy transfer to a partner may be formate dehydrogenase because substrate exchange in form of formate has previously been described to occur between methanogens and sulfur-reducing bacteria[27]. Helarchaeota genomes code for the alpha and beta subunits of a membrane-bound formate dehydrogenase (EC. 1.2.1.2) that could facilitate this transfer (Fig. 2, Supplementary Data 1). However, to our knowledge formate transfer has not been shown to mediate methane oxidation. Alternatively, Helarchaeota may possess a previously undiscovered redox-active complex. In both Helarchaeota bins, a gene cluster was found encoding three proteins that were identified as members of the HydB/Nqo4-like superfamily, Oxidored_q6 superfamily, and a Fe–S disulfide reductase with a FlpD domain (mvhD) (Fig. 5a). An analysis of these three proteins showed that each possessed transmembrane motifs (Fig. 5b, and Supplementary Methods). While the membrane association of the disulfide reductase/FlpD needs to be confirmed, interactions with the other two membrane-associated subunits

may allow for the bifurcated electrons to be transferred across the membrane.

Finally, hydrogen production and release was also considered as a possible electron sink for Helarchaeota. We identified several hydrogenase subunits and putative Fe-S disulfide reductase-encoding genes in the Helarchaeota genomes. Subsequent phylogenetic analyses revealed that the majority of these hydrogenases represent small and large subunits of group IIIC hydrogenases (methanogenic $F_{420}$-non-reducing hydrogenase (*mvh*)) that are usually involved in bifurcating electrons from hydrogen (Supplementary Figure 4, Supplementary Data 1). In contrast, while homologs belonging to the above mentioned Oxidored_q6 superfamily protein family are often found to be associated with group IV hydrogenases, canonical membrane-bound group IV-hydrogenases could not be identified in the genomes of the Helarchaeota. Altogether, this indicates that hydrogen could play a central role in energy metabolism of Helarcharota, but the absence of a classical membrane-bound hydrogenase makes it unlikely that hydrogen is the major syntrophic electron carrier.

## Discussion

Historically methanogenesis and anaerobic methane oxidation were regarded as the only examples of anaerobic archaeal short-chain alkane metabolism. The enzymes acting in these pathways were considered to be biochemically and phylogenetically unique and limited to lineages within the Euryarchaeota[4]. This study represents the discovery of the previously unknown phylum referred to as Helarchaeota, whose members encode a mcr-like gene cluster. This opens the possibility that some representatives of the Asgard archaea may have the potential for anaerobic short-chain alkane oxidation. Since the presence of these *mcr* genes is restricted to Helarchaeota among the known Asgard archaea[19], these genes were likely transferred to Helarchaeota and do not constitute an ancestral trait within the Asgard superphylum. Based on current phylogenetic analysis, the Helarchaeota *mcr* gene cluster may have been horizontally acquired from either Bathyarchaeota or *Ca*. Syntrophoarchaeum (Fig. 1b, Supplementary Figure 3). Due to this close relationship, we based our analysis of Helarchaeota's ability to perform anaerobic short-chain hydrocarbon oxidation on the pathway proposed for Ca. Syntrophoarchaeum. Helarchaeota probably utilize a similar

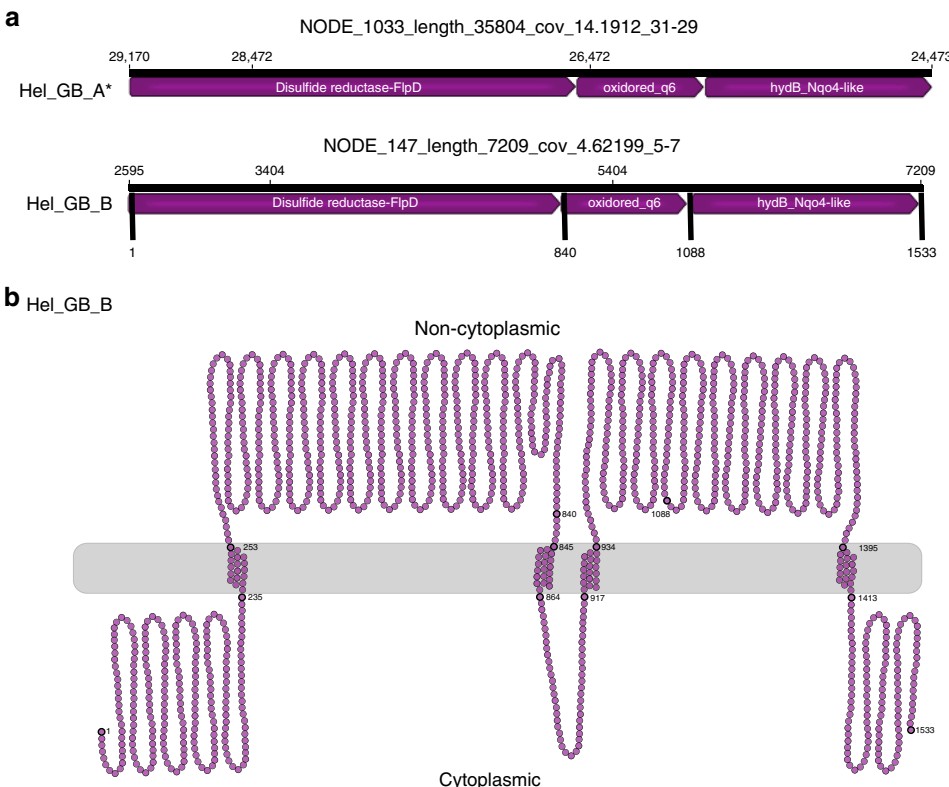

**Fig. 5** Gene cluster found in Helarchaeota that encode for a possible energy-transferring complex. **a** In Hel_GB_A the complex was found on the reverse strand but has been oriented in the forward direction for clarity (asterisk). Arrows indicate the length of the reading frame. Gene names were predicted by various databases (Supplementary Methods). Small numbers located above the arrows refer to the nucleotide position for the full contig. Bold numbers on Hel_GB_B refer to the amino acid number of the whole complex. **b** Figure depicts the membrane motifs identified on NODE_147_length_7209_cov_4.62199_5, 6, and 7 using various programs (Supplementary Methods). Each circle represents a single amino acid. Bold circles represent amino acids at the start of the protein, the start and end of the transmembrane sites, and the end of the complex. Numbering corresponds to the amino acid numbers of Hel_GB_B in panel **a**. A full loop represents 50 amino acids and does not reflect the secondary structure of the complex

short-chain alkane as a substrate in lieu of methane, but given the low-butane concentrations at our site it may not be the only substrate.

Our comparison to *Ca. S. butanivorans* shows a consistent presence in genes necessary for this metabolism including a complete Wood–Ljungdahl pathway, acyl oxidation pathway, and internal electron transferring systems. Some of these electron-transferring systems are essential housekeeping components that may act as electron carriers for oxidation reactions. Interestingly, in the Wood–Ljungdahl pathway identified in *Ca. S. butanivorans*, the bacterial enzyme 5,10-methylene-tetra-hydrofolate reductase (met) is thought to be substituting for the missing 5,10-methylene-tetrahydromethanopterin reductase (mer)[7]. In contrast, Helarchaeota encode the canonical archaeal-type mer. To render anaerobic butane oxidation energetically favorable, it must be coupled to the reduction of an electron acceptor such as nitrate, sulfate or iron[7,26,27]. In ANME archaeum that lack genes for internal electron acceptors, methane oxidation is enabled through the transfer of electrons to a syntrophic partner organism. In Syntrophoarchaeum, syntrophic butane oxidation is thought to occur through the exchange of electrons via pili and/or cytochromes with sulfate-reducing bacteria[7]. Helarchaeota do not appear to encode any of the systems traditionally associated with syntrophy and no partner was identified in this study. Thus, further research is needed to identify possible bacterial partners.

Furthermore, the hypothesis that Helarchaeota have the ability to utilize short-chain alkanes remains to be confirmed as the genomes of members of this group do not encode canonical

routes for electron transfer to a partner bacterium. However, we identified potential enzymes that may be involved in transfer of electrons. Some methanogenic archaea use formate for syntrophic energy transfer to a syntrophic partner; therefore, the reverse reaction has been speculated to be energetically feasible for methane oxidation[27]. If this is true, the presence of a membrane-bound formate dehydrogenase in the Helarchaeota genomes may support this electron-transferring mechanism, however, to our knowledge this has never been shown for an ANME archaea so far. Alternatively, the type 3 NiFe-hydrogenases encoded by Helarchaeota may be involved in transfer of hydrogen to a partner organism. For example, we identified a protein complex distantly related to the *mvh–hdr* of methanogens for electron transfer (Supplementary Methods). *Mvh–hdr* structures have been proposed to be potentially used by facultative hydro-genotrophic methanogens for energy transfer, but the direction-ality of hydrogen exchange could easily be reversed[2]. These methanogens form syntrophic associations with fermenting, $H_2$-producing bacteria, lack dedicated cytochromes or pili and use the *mvh–hdr* for electron bifurcation[2]. The detection of a hydrophobic region in the *mvh–hdr* complex led to the suggestion that this complex could be membrane bound and act as mechanism for electron transfer across the membrane; however, a transmembrane association has never been successfully shown[2]. While the membrane association of the disulfide reductase/FlpD needs to be confirmed, we were able to detect several other transmembrane motifs in the associated proteins that could potentially allow electron transfer in form of hydrogen to an external partner. Thus, while we propose that the most likely

explanation for anaerobic short-chain alkane oxidation in Helarchaeota is via a syntrophic interaction with a partner, additional experiments are needed to confirm this working hypothesis.

The discovery of alkane-oxidizing pathways and possible syntrophic interactions in a phylum of Asgard archaea indicates a much wider phylogenetic range for hydrocarbon utilization. Based on phylogenetic analyses it seems most likely that the Helarchaeota *mcr* operon may have been horizontally transferred from either Bathyarchaeota or Syntrophoarchaea. However, the preservation of a horizontally transferred pathway is indicative of a competitive advantage; it follows that gene transfers among different archaeal phyla reflect alkane oxidation as a desirable metabolic trait. The discovery of the alkyl-CoM reductases and alkane-oxidizing pathways among the Asgard archaea indicates ecological roles for these still cryptic organisms, and opens up a wider perspective on the evolution and expansion of hydrocarbon-oxidizing pathways throughout the archaeal domain.

## Methods

**Sample collection and processing**. Samples analyzed here are part of a study that aimed to characterize the geochemical conditions and microbial community of GB hydrothermal vent sediments (Gulf of California, Mexico)[31,32]. The two genomic bins discussed in this paper, Hel_GB_A and Hel_GB_B, were obtained from sediment core samples collected in December 2009 on *Alvin* dives 4569_2 and 4571_4, respectively[21]. Immediately after the dive, freshly recovered sediment cores were separated into shallow (0–3 cm), intermediate (12–15 cm), and deep (21–24 cm) sections for further molecular and geochemical analysis, and frozen at −80 °C on the ship until shore-based DNA extraction. Hel_GB_A was recovered from the intermediate sediment (~28 °C) and Hel_GB_B was recovered from shallow sediment (~10 °C) from a nearby core (Supplementary Table 1); the sampling context and geochemical gradients of these hydrothermally influenced sediments are published and described in detail[21,31].

DNA was extracted from sediment samples using the MO BIO—PowerMax Soil DNA Isolation kit and sent to the Joint Genome Institute (JGI) for sequencing.

**JGI generation of reads and processing of data**. A half a lane of Illumina reads (HiSeq-2500 1TB, read length of $2 \times 151$ bp) were generated at Joint Genome Institute for each sample, producing a total of 226,647,966 and 241,605,888 reads from dives 4569-2 and 4571-4, respectively. The average percent of reads with a phred-score ($Q$) ≥ 30 was 86.2% and 90.39% and the average base quality score was $34.35 \pm 7.73$ and $35.38 \pm 6.52$ for samples from dive 4569-2 and 4571-4, respectively. The JGI performed read quality checks and generated a first assembly using the following methods: BBDuk adapter trimming removed known Illumina adapters. The reads were further processed using BBDuk quality filtering and trimming to remove reads with a quality score less than 12, containing more than three "Ns", or with quality scores (before trimming) averaging less than 3 over the read length, or length under 51 bp after trimming. In addition, reads matching Illumina artifacts or phiX were discarded. The remaining reads were mapped to a masked version of the human HG19 with BBMap and all hits over 93% sequence identity to the human genome were discarded. Trimmed, screened, paired-end Illumina reads were assembled using the megahit assembler using a range of kmers. Assemblies were preformed with default parameters in megahit with the following options: "–k-list 23,43,63,83,103,123". High-quality reads were mapped to the final assembly to calculate coverage information using bbmap by excluding all parameters except ambiguous=random as described by JGI.

**Genome reconstruction**. The contigs from the JGI assembled data were binned using ESOM[33], MetaBAT[34], and CONCOCT[35] and resulting bins were combined using DAS Tool (version 1.0)[36]. For ESOM, binning was performed on contigs with a minimum length of 2000 bp using the K-batch algorithm for training after running the perl script esomWrapper.pl[33]. Emerging self-organizing maps (ESOM) were manually sorted and curated. The bins were extracted using getClassFasta.pl (using −loyal 51). Reference genomes were included to add genetic signatures for the assembled contigs and improve binning. For CONCOCT, Anvi'o (v2.2.2) was used as the metagenomic workflow pipeline[37]. Coverage information was obtained by mapping all high-quality reads of each sample against the assembly of another sample using the BWA-MEM algorithm in paired-end mode (bwa-0.7.12-r1034; using default settings)[38]. The resulting sam file was sorted and converted to bam using samtools (version 0.1.19)[39]. The bam file was prepared for Anvi'o using the script anvi-init-bam and a contigs database generated using anvi-gen-contigs-database. These files were the input for anvi-profile. Generated profiles for the assemblies were combined using anvi-merge and the resulting bins summarized using anvi-summarize (-C CONCOT)[37]. If not mentioned otherwise, the scripts

were used with default settings. Metabat was also used as a binning approach (v1)[34]. As described for Anvi'o the input consisted of the scaffold files (≥2000 bp) and the mapping files. First, each of the mapping files were summarized using jgi_summarize_bam_contig_depths and then metabat was run using the following settings: –minProb 75 –minContig 2000 –minContigByCorr 2000. Results from the three different binning tools were combined using DAS Tool (version 1.0)[36]. For each of the binning tools a scaffold-to-bin list was prepared and DAS Tool run on each of the eleven scaffold files as follows: DAS_Tool.sh -i Anvio_contig_list.tsv, Metabat_contig_list.tsv,ESOM_contig_list.tsv -l Anvio,Metabat,ESOM -c scaffolds. fasta –write_bins 1. CheckM lineage_wf (v1.0.5) was run on bins generated from DAS_Tool and 577 bins showed an completeness >50% and were characterized further[40]. 37 Phylosift[41] identified marker genes were used for preliminary phylogenetic identification of individual bins (Supplementary Table 5). Thereby, we identified two genomes, belonging to a previously uncharacterized phylum within the Asgard archaea, which we named Helarchaeota. To improve the quality of the two Helarchaeota genomes IDBA-UD was run on raw data using the command: "idba_ud -r Guay9_METAGENOME.fasta -o G9 –pre_correction –mink 75 –maxk 105 –seed_kmer 55 –num_threads 30". Metaspades was run on Raw data and Metabat assembled bins using as follows: "metaspades.py –12 Guay16.11400.5.204 846.CTCTCTA-CGTCTAA.filter-METAGENOME.fastq -o Metaspades –only-assembler –meta". Binning procedures (using scaffolds longer than 2000 bp) as described above for the original bins were repeated with these redone assemblies. All bins were compared to the original Helarchaeota bins using blastn[42] for identification. Mmgenome[43] and CheckM[40] were used to calculate genome statistics (i.e., contig length, genome size, contamination, and completeness). The highest quality Helarchaeota bin from each sample was chosen for further analyses. For the 4572-4 dataset, the best bin was generated using the Metaspades reassembly on the trimmed data and for the 4569-2 dataset the best bin was recovered using the Metaspades reassembly on the original Hel bin contigs. The final genomes were further cleaned by GC content, paired-end connections, sequence depth and coverage using Mmgenome[43]. CheckM was rerun on cleaned bins to estimate the Hel_GB_A to be 82% complete and Hel_GB_B to be 87% complete and both bins were characterized by a low degree of contamination (between 1.4 and 2.8% with no redundancy) (Table 1)[40]. Genome size was estimated to be 4.6 Mbp for Hel_GB_A and 4.1 for Hel_GB_B and was calculated using percent completeness and bin size to extrapolate the likely size of the complete genome. CompareM was used to analysis differences between Helarchaeota bins and published Asgard bins using the command python comparem aai_wf –tmp_dir tmp/ –file_ext fa -c 8 aai_-compair_loki aai_compair_loki_output (https://github.com/dparks1134/CompareM). Read abundance summarized by jgi_summarize_bam_contig_depths were used to calculate relative read abundance and total percent of metagenomic reads. Relative read abundance was calculated as total read abundance normalized to genome size and divided by total reads. Relative read abundance was then multiplied by the constant $1 \times 10^{12}$ for clarity. Total percent of metagenomic reads was calculated as total read abundance divided by total reads times 100. Relative read abundance was compared to other genomics bins recovered from these sites to look for co-occurrence[20].

**16S rRNA gene analysis**. Neither bin possessed a 16S rRNA gene sequence[41], and to uncover potentially unbinned 16S rRNA gene sequences from Helarchaeota, all 16S rRNA gene sequences obtained from samples 4569_2 and 4571_4 were identified using JGI-IMG annotations, regardless of whether or not the contig was successfully binned. These 16S rRNA gene sequences were compared using blastn[42] (blastn -outfmt 6 -query Hel_possible_16s.fasta –db Hel_16s -out Hel_p-ossible_16s_blast.txt -evalue 1E-20) to recently acquired 16S rRNA gene sequences from MAGs recovered from preliminary data from additional GB sites. A 37 Phylosift[41] marker genes tree was used to assign taxonomy to these MAGs. We were able to identify five MAGs that possessed 16S and that formed a monophyletic group with our Hel_GB bins (Supplementary Table 2; Megxx in Fig. 2). Of the unbinned 16S rRNA gene sequences one was identified as likely Helarchaeota sequence. The contig was retrieved from the 4572_4 assembly (designated Ga0180301_10078946) and was 2090 bp long and encoded for an 16S rRNA gene sequence that was 1058 bp long. Given the small size of this contig relative to the length of the 16S rRNA gene none of the other genes on the contig could be annotated. Blastn[42] comparison to published Asgard 16S rRNA gene sequences was performed using the following command: blastn -outfmt 6 -query Hel_p-ossible_16s.fasta –db Asgrad_16s -out Hel_possible_16s_blast.txt -evalue 1E-20 (Supplementary Table 2). The GC content of each 16S rRNA gene sequence was calculated using the Geo-omics script length+GC.pl (https://github.com/Geo-omics/scripts/blob/master/AssemblyTools/length%2BGC.pl). For a further phylogenetic placement, the 16S rRNA gene sequences were aligned to the SILVA database (SINA v1.2.11) using the SILVA online server[44] and Geneious (v10.1.3)[45] was used to manually trim sequences. The alignment also contained 16S rRNA gene sequences from the preliminary Helarchaeota bins. The cleaned alignment was used to generated a maximum-likelihood tree with RAxML as follows: "/raxmlHPC-PTHREADS-AVX -T 20 -f a -m GTRGAMMA -N autoMRE -p 12345 -x 12345 -s Nucleotide_alignment.phy -n output" (Fig. 1b).

**Phylogenetic analysis of ribosomal proteins**. For a more detailed phylogenetic placement, we used BLASTp[46] to identify orthologs of 56 ribosomal proteins in the

two Helarchaeota bins, as well as from a selection of 130 representative taxa of archaeal diversity and 14 eukaryotes. The full list of marker genes selected for phylogenomic analyses is shown in Supplementary Table 6. Individual protein datasets were aligned using mafft-linsi[47] and ambiguously aligned positions were trimmed using BMGE (-m BLOSUM30)[48]. Maximum likelihood (ML) individual phylogenies were reconstructed using IQtree v. 1.5.5[49] under the LG+C20+G substitution model with 1000 ultrafast bootstraps that were manually inspected. Trimmed alignments were concatenated into a supermatrix, and two additional datasets were generated by removing eukaryotic and/or DPANN homologs to test the impact of taxon sampling on phylogenetic reconstruction. For each of these concatenated datasets, phylogenies were inferred using ML and Bayesian approaches. ML phylogenies were reconstructed using IQtree under the LG+C60+F+G +PMSF model[50]. Statistical support for branches was calculated using 100 boot-straps replicated under the same model. To test robustness of the phylogenies, the dataset was subjected to several treatments. For the "full dataset" (i.e., with all 146 taxa), we tested the impact of removing the 25% fastest-evolving sites, as within a deep phylogenetic analysis, these sites are often saturated with multiple substitutions and, as a result of model-misspecification can manifest in an artifactual signal[51–53]. The corresponding ML tree was inferred as described above. Bayesian phylogenies were reconstructed with Phylobayes for the dataset "without DPANN" under the LG+GTR model. Four independent Markov chain Monte Carlo were run for ~38,000 generations. After a burn-in of 20%, convergence was achieved for three of the chains (maxdiff < 0.29). The initial supermatrix was also recoded into four categories, in order to ameliorate effects of model misspecification and saturation[54] and the corresponding phylogeny was reconstructed with Phylobayes, under the CAT+GTR model. Four independent Markov chain Monte Carlo chains were run for ~49,000 generations. After a burn-in of 20 convergence was achieved for all four chains (maxdiff < 0.19). All phylogenetic analyses performed are summarized in Supplementary Table 7, including maxdiff values and statistical support for the placement of Helarchaeota, and of eukaryotes.

**Phylogenetic analysis of McrA and concatenated McrAB.** McrA homologs were aligned using mafft-linsi[47], trimmed with trimAL[55], and the final alignment consisting of 528 sites was subjected to phylogenetic analyses using IQtree v. 1.5.5[49] with the LG+C60+R+F model. Support values were estimated using 1000 ultrafast boostraps[56] and SH-like approximate likelihood ratio test[57], respectively. Sequences for McrA and B were aligned separately with mafft-linsi[47] and trimmed using trimAL Subsequently, McrA and McrB encoded in the same gene cluster, were concatenated yielding a total alignment of 972 sites. Bayesian and ML phylogenies were inferred using IQtree v. 1.5.5[49] with the mixture model LG+C60+R+F and PhyloBayes v. 3.2[58] using the CAT-GTR model. For ML inference, support values were estimated using 1000 ultrafast boostraps[56] and SH-like approximate like-lihood ratio test[57], respectively. For Bayesian analyses, four chains were run in parallel, sampling every 50 points until convergence was reached (maximum dif-ference < 0.07; mean difference < 0.002). The first 25% or the respective generations were selected as burn-in. Phylobayes posterior predictive values were mapped onto the IQtree using sumlabels from the DendroPy package[59]. The final trees were rooted artificially between the canonical Mcr and divergent Mcr-like proteins, respectively. Original alignment and treefiles are available upon request.

**Metabolic analyses.** Gene prediction for the two Helarchaeota bins was per-formed using prodigal[60] (V2.6.2) with default settings and Prokka[61] (v1.12) with the extension "–kingdom archaea". Results for both methods were comparable and yielded a total of 3574–3769 and 3164–3287 genes for Hel_GB_A and Hel_GB_B, respectively, with Prokka consistently identifying fewer genes. Genes were anno-tated by uploading the protein fasta files from both methods to KAAS (KEGG Automatic Annotation Server) for complete or draft genomes to assign orthologs[62]. Files were run using the following settings: prokaryotic option, GhostX, and bi-directional best hit (BBH)[62]. Additionally, genes were annotated by JGI-IMG[63] to confirm hits using two independent databases. Hits of interest were confirmed using blastp on the NCBI webserver[46]. The dbCAN[64] and MEROPS[65] webserver were run using default conditions for identification of carbohydrate degrading enzymes and peptidases respectively. Hits with e-values lower than $e^{-20}$ were discarded. In addition to these methods an extended search was used to categorize genes involved in butane metabolism, syntrophy and energy transfer.

Identified genes predicted to code for putative alkane oxidation proteins were similar to those described from Candidatus Syntrophoarchaeum spp. Therefore, a blastp[46] database consisting of proteins predicted to be involved in the alkane oxidation pathway of Ca. Syntrophoarchaeum was created in order to identify additional proteins in Helarchaeota, which may function in alkane oxidation. Positive hits were confirmed with blastp[46] on the NCBI webserver and compared to the annotations from JGI-IMG[63], Interpro[66], Prokka[61], and KAAS[62] annotation. Genes for mcrABG were further confirmed by a HMMER[67] search to a published database using the designated threshold values[68] and multiple MCR trees (see Methods). To confirm that the contigs with the mcrA gene cluster were not missbinned, all other genes on these contigs were analyzed for their phylogenetic placement and gene content. The prodigal protein predictions for genes on the contigs with mcrA operons were used to determine directionality and length of the potential operon.

To identify genes that are involved in electron and hydrogen transfer across the membrane, a database was created of known genes relevant in syntrophy that were

download from NCBI. The protein sequences of the two Helarchaeota genomes were blasted against the database to detect relevant hits (E-value ≥ $e^{-10}$). All hits were confirmed using the NCBI webserver, Interpro, JGI-IMG, and KEGG. Hydrogenases were identified by a HMMER search to published database using the designated threshold values. Hits were confirmed with comparisons against JGI annotations and NCBI blasts, the HydDB database[69] and a manual database made from published sequences[70,71]. All detected hydrogenases were used to generate two phylogenetic trees, one for proteins identified as small subunits and one for large subunits in order to properly identify the different hydrogenase subgroups. Hydrogenases that are part of the proposed complex were then further analyzed to evaluate if this was a possible operon by looking for possible transcription factors and binding motifs (Supplementary Methods).

**ESP identification.** Gene prediction for the two Helarchaeota bins was performed using prodigal[60] (V2.6.2) with default settings. All the hypothetical proteins inferred in both Helarchaeaota were used as seeds against InterPro[66], arCOG[72], and nr using BLAST[46]. The annotation table from Zaremba-Niedzwiedzka et al.[12] was used as a basis for the comparison. The IPRs (or in some cases, the arCOGs) listed in the Zaremba-Niedzwiedzka et al. were searched for in the Helarchaeota genomes[12], and the resulting information was used to complete the presence/absence of table. When something that had previously been detected in an Asgard bin was not found in a Helarchaeota bin using the InterPro/arCOG annotations, BLASTs were carried out using the closest Asgard seeds to verify the absence. In some cases, specific analyses were used to verify the homology or relevance of particular sequences. The details for each individual ESP are depicted in Supplementary Methods.

**Reporting Summary.** Further information on experimental design is available in the Nature Research Reporting Summary linked to this article.

## Data availability

The raw reads from the metagenomes described in this study are available at JGI under the IMG genome IDs 3300014911 and 3300013103 for samples 4569-2 and 4571-4, respectively. Genome sequences are available at NCBI under the Accession numbers SAMN09406154 and SAMN09406174 for Hel_GB_A and Hel _GB_B, respectively. Both are associated with BioProject PRJNA362212.

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

## Acknowledgements

This study was supported in part by an Alfred P. Sloan Foundation fellowship (FG-2016-6301) and National Science Foundation Directorate of Biological Sciences (Systematics and Biodiversity Sciences) (Award 1737298) to B.J.B. Sampling in Guaymas Basin and post-cruise work was supported by NSF Awards OCE-0647633 and OCE-1357238 to APT, respectively. The work conducted by the U.S. Department of Energy Joint Genome Institute, a DOE Office of Science User Facility, is supported by the Office of Science of the U.S. Department of Energy under Contract No. DE-AC02-05CH11231 provided to N.D. A.S. was supported by a Marie Curie IEF European grant (625521), a VR starting grant (2016-03559) and a WISE fellowship by the NWO-I Foundation of the Netherlands Organisation for Scientific Research. L.E. was funded by the European Union's Horizon 2020 research and innovation programme under the Marie Sklodowska-Curie grant agreement No 704263. This work was supported by grants of the European Research Council (ERC Starting grant 310039-PUZZLE_CELL), the Swedish Foundation for Strategic Research (SSF-FFL5) and the Swedish Research Council (VR grant 2015-04959) to T.J.G.E.

## Author contributions

K.W.S., T.J.G.E., N.D. and B.J.B. conceived the study. K.W.S., N.D. and B.J.B. analyzed the genomic data. A.P.T. collected and processed the samples. K.W.S., A.S. and L.E. performed the phylogenetic analyses. J.L. analyzed the ESPs. K.W.S., J.R.S., A.P.T. and B.J.B. handled the metabolic inferences. B.J.B. and K.W.S. wrote the paper with inputs from all authors.

## Additional information

**Competing interests:** The authors declare no competing interests.

