## [Peer Review File · Nature Communications]

Editorial Note: This manuscript has been previously reviewed at another journal that is not operating a transparent peer review scheme. This document only contains reviewer comments and rebuttal letters for versions considered at *Nature Communications*. Mentions of the other journal have been redacted.

Reviewers' comments:

Reviewer #1 (Remarks to the Author):

Seitz et al present a revised version of their paper entitled "New Asgard Archaea capable of anaerobic hydrocarbon cycling", which was originally submitted to [redacted]. Overall, the authors responses to my questions were satisfying. I particularly like that they included an analysis of the active enter of the respective enzyme. The new version of the manuscript is technically sound, their conclusions are justified and I thus only have two minor comments on the manuscript. Overall I congratulate the authors on this fine piece of science.

1. I think that the term "new Asgard archaea" in the title of the manuscript (and elsewhere) is misleading. The term new, could relate to either their discovery or to their appearance in earth history (from an evolutionary perspective). I suggest avoiding terms like "new" or "novel" to avoid this ambiguity. I'm aware that I did not make this comment on the original version and I apologize for that.

2. Jan 19 2019 a paper appeared about divergent *mcrA* genes discovered in the seafloor. I would like to request that the authors include the sequences from this recent publication in their phylogenetic analyses to make sure that their claims are not threatened by these new sequences. This could be a supplemental figure.

Reviewer #3 (Remarks to the Author):

The authors have done a reasonable job responding to my concerns.

I would still like to know the 16S rRNA sequence identity between the Helarchaeota and the other archaeal phyla. I appreciate the inclusion of the 16S rRNA tree, but think identity remains an important factor to report (though I appreciate phylum-level 16S rRNA identity boundaries are blurrier than other taxonomic levels).

Reviewers' comments:

Reviewer #1 (Remarks to the Author):

1. I think that the term “new Asgard archaea” in the title of the manuscript (and elsewhere) is misleading. The term new, could relate to either their discovery or to their appearance in earth history (from an evolutionary perspective). I suggest avoiding terms like “new” or “novel” to avoid this ambiguity. I’m aware that I did not make this comment on the original version and I apologize for that.

We understand the Reviewer’s call for clarity therefore the words “new” and “novel” have been replaced with terms such as “previously unknown” and “newly discovered” to remove uncertainty.

2. Jan 19 2019 a paper appeared about divergent mcrA genes discovered in the seafloor. I would like to request that the authors include the sequences from this recent publication in their phylogenetic analyses to make sure that their claims are not threatened by these new sequences. This could be a supplemental figure.

At the request of the Reviewer we have replaced Supplementary Figure 2 with a version that contains the McrA sequences from Boyd et al., 2019. The tree was run under identical conditions as before with the only change being the additional sequences.

Reviewer #3 (Remarks to the Author):

The authors have done a reasonable job responding to my concerns. I would still like to know the 16S rRNA sequence identity between the Helarchaeota and the other archaeal phyla. I appreciate the inclusion of the 16S rRNA tree, but think identity remains an important factor to report (though I appreciate phylum-level 16S rRNA identity boundaries are blurrier than other taxonomic levels).

At the Reviewer’s request a direct statement comparing Helarchaeota’s 16S rRNA percent identity to Lokiarchaeum GC14_75. Comparison to other Asgard 16S rRNA sequences can be found in (Supplementary Table 2).